# Captive chimpanzee foraging in a social setting: a test of problem solving, flexibility, and spatial discounting

Lydia M. Hopper[1], Laura M. Kurtycz[1], Stephen R. Ross[1] and Kristin E. Bonnie[1,2]

[1] Lester E. Fisher Center for the Study & Conservation of Apes, Lincoln Park Zoo, Chicago, IL, USA
[2] Department of Psychology, Beloit College, Beloit, WI, USA

## ABSTRACT

In the wild, primates are selective over the routes that they take when foraging and seek out preferred or ephemeral food. Given this, we tested how a group of captive chimpanzees weighed the relative benefits and costs of foraging for food in their environment when a less-preferred food could be obtained with less effort than a more-preferred food. In this study, a social group of six zoo-housed chimpanzees (*Pan troglodytes*) could collect PVC tokens and exchange them with researchers for food rewards at one of two locations. Food preference tests had revealed that, for these chimpanzees, grapes were a highly-preferred food while carrot pieces were a less-preferred food. The chimpanzees were tested in three phases, each comprised of 30 thirty-minute sessions. In phases 1 and 3, if the chimpanzees exchanged a token at the location they collected them they received a carrot piece (no travel) or they could travel ≥10 m to exchange tokens for grapes at a second location. In phase 2, the chimpanzees had to travel for both rewards (≥10 m for carrot pieces, ≥15 m for grapes). The chimpanzees learned how to exchange tokens for food rewards, but there was individual variation in the time it took for them to make their first exchange and to discover the different exchange locations. Once all the chimpanzees were proficient at exchanging tokens, they exchanged more tokens for grapes (phase 3). However, when travel was required for both rewards (phase 2), the chimpanzees were less likely to work for either reward. Aside from the alpha male, all chimpanzees exchanged tokens for both reward types, demonstrating their ability to explore the available options. Contrary to our predictions, low-ranked individuals made more exchanges than high-ranked individuals, most likely because, in this protocol, chimpanzees could not monopolize the tokens or access to exchange locations. Although the chimpanzees showed a preference for exchanging tokens for their more-preferred food, they appeared to develop strategies to reduce the cost associated with obtaining the grapes, including scrounging rewards and tokens from group mates and carrying more than one token when travelling to the farther exchange location. By testing the chimpanzees in their social group we were able to tease apart the social and individual influences on their decision making and the interplay with the physical demands of the task, which revealed that the chimpanzees were willing to travel farther for better.

Corresponding author
Lydia M. Hopper,
lhopper@lpzoo.org

## INTRODUCTION

How, when, where, and for how long animals forage for food is influenced by external factors (e.g., predation risk, social interactions, and prey availability), internal factors (e.g., hunger, food preferences, and animals' age or sex), and phylogenetic factors (e.g., physiological parameters and sensory limitations) (*Pianka, 1997*). The optimal foraging theory proposes that foraging should increase fitness while reducing foraging costs (*Pyke, 1984*; *Bautista, Tinbergen & Kacelnik, 2001*) and suggests that animals should prefer to travel a shorter, rather than a longer, distance to obtain food if all other options are equal (*Blaser & Ginchansky, 2012*; *Reilly et al., 2012*). However, if the food that is farther away is 'better' (e.g., larger, more plentiful, more preferred) than the closer option, will animals exert more effort to obtain the better food? Humans, for example, will travel farther to reach their more-preferred restaurants (*Froehlich et al., 2006*), and we show such behavioral choices in other contexts too, including travelling farther to visit more closely related kin (*Pollet, Roberts & Dunbar, 2013*) and to reach hospitals that offer higher quality healthcare (*Romley & Goldman, 2011*). Like humans, nonhuman primates not only have individual preferences for certain foods, but seek them out within their environment (*Janmaat, Byrne & Zuberbühler, 2006*; *Janmaat, Ban & Boesch, 2013*), suggesting that they would be likely to travel farther for better.

In the natural habitat of primates, food sources are not evenly distributed, either physically or temporally. In order to obtain preferred or ephemeral foods, primates are selective about which trees they feed from (*Glander, 1979*) and the routes that they travel to reach them (*Ban, Boesch & Janmaat, 2014*). Primates also appear willing to travel farther to obtain their more-preferred foods, even if less-preferred foods are closer or easier to obtain. For example, observations of wild black howler monkeys (*Alouatta pigra*) suggests that their foraging strategy is influenced by both distance (they preferentially travel to closer trees) and food desirability (they use step-wise movements to reach high quality patches, (*Plante, Colchero & Calmé, 2014*)). Similarly, wild capuchins (*Cebus apella nigritus*) have been shown to weigh the relative cost of distance and food availability such that they will detour from a direct path to food only when it enables them to gain additional, and worthwhile, food items (*Janson, 2007*). Female chimpanzees (*Pan troglodytes*) have been reported to leave their nighttime nests earlier when more-preferred fig trees are located farther away (*Janmaat et al., 2014*), indicating that they are willing to travel longer distances to reach more desirable food rewards and that they plan their activities in order to reach them. Even in a captive setting, where primates are not required to forage for their daily food to survive, they have been reported to use efficient routes when searching for foods, sometimes bypassing less-preferred foods to reach more-preferred foods first (e.g., *Menzel, 1973*; *Boesch & Boesch, 1984* provides a review), but evidence for such strategic foraging is mixed (e.g., *Howard & Fragaszy, 2014*).

Many primates live in social groups and so an individual's foraging strategy and food choices may also be influenced by the decisions and preferences of its group mates as well as the effort required to obtain that food (*Finestone et al., 2014*; *Hardus et al., 2015*; *Marshall et al., 2015*). Indeed, an advantage of social living for gregarious primate species

is that they can use social information to learn where to find food (*Rapaport & Brown, 2008*), which foods to eat (*Visalberghi & Addessi, 2000*; *van de Waal, Borgeaud & Whiten, 2013*), and how to process those foods (*Boinski & Timm, 1985*; *van de Waal, Bshary & Whiten, 2014*). In experimental tests, chimpanzees have been shown to copy the foraging techniques of their group mates (e.g., *Whiten, Horner & de Waal, 2005*), even when the technique appears arbitrary (*Bonnie et al., 2007*). Furthermore, tests with socially-housed captive chimpanzees have identified that low-ranking and 'uncertain' chimpanzees appear more likely than dominants to use social information (*Kendal et al., 2015*), and they preferentially copy dominant or 'expert' individuals (*Horner et al., 2010*), even when doing so means they received a less-preferred food reward (*Hopper et al., 2011*). Therefore, although tests with individual primates can reveal how they weigh the physical costs of foraging (e.g., *Stevens et al., 2005*), studies run with socially-housed captive primates can also uncover how they weigh the social costs of foraging along with the physical costs.

Previous tests of socially-housed captive primates' foraging have assessed their ability to learn novel extractive foraging tasks from observing others (e.g., *P. troglodytes*, *Whiten, Horner & de Waal, 2005*; *Saimiri boliviensis, Hopper et al., 2013*), but only a few have offered differentially-valued food rewards in such research paradigms (e.g., *Dean et al., 2012*). Two such studies (*Hopper et al., 2011*; *Van Leeuwen et al., 2013*) presented chimpanzees with opportunities to exchange tokens for food rewards in a group setting, a technique most notably utilized in tests of social learning (e.g., *Bonnie et al., 2007*; *Horner et al., 2010*). In these two experiments, chimpanzees were given the option to exchange tokens for two differently-valued foods, revealing that they could discriminate between the two options (*Hopper et al., 2011*; *Van Leeuwen et al., 2013*). Unlike in the examples of wild primate foraging described above, in these two studies the effort required to obtain both rewards was the same, even though they were valued differently by the chimpanzees. In *Hopper et al.*'s (*2011*) study, chimpanzees could exchange different colored tokens that were assigned different reward values. As the chimpanzees collected both token types at the same location and exchanged them all with a researcher at a second location 8 m away, the effort required was equal irrespective of which color token the chimpanzees exchanged and, therefore, which reward they received (less-preferred carrot piece or more-preferred grape). Similarly, in *Van Leeuwen et al.*'s (*2013*) study, chimpanzees could collect tokens from within their enclosure and then trade them at one of two locations, one where they received a high-value reward (5 peanuts) or one where they received a low-value reward (1 peanut). However, as the distance between the two exchange locations and the token collection point was the same (30 m), the effort required to obtain either reward value was also the same.

Although not designed to test the impact of the physical environment on the chimpanzees' food choices, these two token exchange studies (*Hopper et al., 2011*; *Van Leeuwen et al., 2013*) did highlight the chimpanzees' flexible foraging strategies. In *Hopper et al.*'s (*2011*) study, because the chimpanzees exchanged their tokens with a researcher at a single location, subordinate chimpanzees could not easily avoid competition and often had tokens taken from them by their group mates (especially those that garnered

the high-value grapes). Although all the chimpanzees had shown an individual preference for one food type (grapes), certain low-ranking chimpanzees often exchanged the tokens that were worth the less-preferred rewards (carrot pieces). *Hopper et al. (2011)* concluded that the chimpanzees switched away from their preferred technique in order to ameliorate competition while still obtaining food rewards, even if they were the less-preferred option. Flexibility of a different kind was revealed by *Van Leeuwen et al.*'s (*2013*) study. Despite having previous personal experience of exchanging tokens for food rewards at a certain location, when a second exchange location became more profitable, the chimpanzees switched strategies and exchanged tokens at the new location where they could obtain more rewards for each exchange. The chimpanzees preferentially exchanged tokens where they could obtain more rewards, even though they had to switch from a previously-learned strategy, something it has been suggested that primates fail to do when both options are equal (e.g., *C. apella*, *Brosnan & de Waal, 2004*; *P. troglodytes*, *Hrubesch, Preuschoft & van Schaik, 2009*).

Extending upon these token exchange methods, and to test whether a group of captive chimpanzees would choose to travel farther to obtain a more-preferred reward, we provided a group of chimpanzees with a single location where they could collect PVC tokens and two locations where they could exchange the tokens with researchers for differentially-valued foods. Crucially, the distance between the token collection point and the two exchange locations was different, such that the chimpanzees had to travel farther to obtain their more-preferred food. As we also wanted to test whether they would be able to shift their foraging locations (c.f., *Van Leeuwen et al., 2013*) we changed the locations where they could exchange their tokens over the course of this 15 month study.

We had a number of key aims with this study, all of which related to how the individual chimpanzees foraged in a dynamic social and physical environment. Accordingly, we also had a number of predictions. Chimpanzees can learn how to exchange tokens very quickly when given training (e.g., *Hopper et al., 2011*; *Van Leeuwen et al., 2013*) or from watching group mates (*Bonnie et al., 2007*). Beyond this, in previous studies that have presented socially-housed primates opportunities to exchange tokens for food rewards, dominant individuals monopolized access to tokens (*P. troglodytes*, *Bonnie et al., 2007*) and made more exchanges with them (*C. apella*, *Addessi et al., 2011*).Therefore, our first prediction was that all the chimpanzees would learn how to exchange the tokens with researchers for food rewards, either through individual trial-and-error learning or by using social information, and that, once the chimpanzees had learned to exchange tokens for food, higher-ranking chimpanzees would exchange more tokens than low-ranking individuals. Accordingly, the chimpanzees received no training prior to the start of this study either in how to exchange tokens or where the exchange locations were.

Previous research has demonstrated that primates can associate different types of tokens with different associated food rewards (*C. apella*, *Brosnan & de Waal, 2004*; *Addessi, Crescimbene & Visalberghi, 2007*; *Addessi et al., 2011*; *P. troglodytes*, *Hopper et al., 2011*; *Van Leeuwen et al., 2013*) and that chimpanzees will selectively exchange tokens at specific locations in order to obtain more desirable rewards when the two options are equally acces-

sible (*Van Leeuwen et al., 2013*). Extending this, we were interested in the likelihood of the chimpanzees traveling farther for the more-preferred rewards and if they would continue to do so when the locations where they could exchange tokens for food rewards changed. Our second prediction, therefore, was that the chimpanzees would exchange more tokens for their more preferred foods, even though they had to travel farther to do so. However, as previously comparable studies have shown that chimpanzees attempt to obtain tokens or rewards opportunistically, by scrounging them from their group mates (*Hopper et al., 2011*; *Van Leeuwen et al., 2013*), our third prediction was that the chimpanzees would scrounge from one another and that dominant individuals would be more likely than subordinates to scrounge tokens and food rewards from their group mates.

## MATERIALS AND METHODS

### Subjects and housing

This study was conducted with six zoo-born, mother-reared chimpanzees (two males, four females) housed together in one social group at the Lincoln Park Zoo, Chicago, USA. At the start of the 15-month study, the average age of the chimpanzees was 19.8 years (range = 13.2–28.4 years). The chimpanzees participate in regular cognitive testing in a group setting using touchscreen interfaces (*Wagner & Ross, 2013*) and their tool-use behavior has also been extensively studied (e.g., *Lonsdorf et al., 2009*; *Bonnie, Ross & Lonsdorf, 2012*; *Calcutt et al., 2014*; *Hopper et al., 2015*). However, at the start of this study, the chimpanzees were naïve to exchanging PVC tokens with researchers in order to obtain food rewards and the two researchers who ran the experimental sessions (LMH and LMK) had not previously worked with this particular group of chimpanzees. Therefore, the chimpanzees had no prior experience receiving food rewards from these researchers or exchanging the PVC tokens with animal care staff or researchers for food rewards.

The chimpanzee group was housed in an expansive indoor-outdoor enclosure at the Regenstein Center for African Apes at Lincoln Park Zoo, which incorporated climbing structures, deep-mulch bedding and an off-exhibit holding area. The indoor exhibit measured 408 m$^2$ and the outdoor exhibit measured 2011 m$^2$ (total = 2419 m$^2$). Throughout the study, the chimpanzees had outdoor access when weather conditions were appropriate ($>5$ °C) but all testing was conducted in their indoor enclosure. In addition to any food rewards that the chimpanzees obtained during the course of this study, fresh produce and primate chow were scattered twice daily throughout their exhibits. Zoo guests were able to observe every test session, and trained educators interpreted every test session to communicate the importance of cognitive testing with the animals.

### Ethics

This study was approved by the Lincoln Park Zoo Research Committee, which is the governing body for all animal research at the institution. No social group manipulations occurred as the result of this project. Food substances, amount, and frequency were reviewed and approved by veterinary and nutrition staff prior to the start of the project. No modifications were made to standard animal care routines. This research adhered

to legal requirements in the United States of America and to the American Society of Primatologists' Principles for the Ethical Treatment of Nonhuman Primates.

## Food preference testing

Before testing began, food preference tests were run with the six chimpanzees to determine their more-preferred and less-preferred foods. As chimpanzee food choices are known to be affected by social influences (*Finestone et al., 2014*), and because we tested the chimpanzees in a group setting, we tested each chimpanzee's individual food preferences while in the presence of their group mates, (c.f., *Hopper et al., 2011*). Food preferences were determined using a forced-choice paradigm (c.f., *Hopper et al., 2014a*). Chimpanzees were each offered two different food items presented on long wooden skewers held at shoulder width by a member of animal care staff. As soon as a chimpanzee selected one of the two food items, by reaching through the cage mesh with their fingers or mouth, the member of animal care staff gave them that food item and withdrew the other option. Ten such tests were run on one day and ten more on a second day with each of the six chimpanzees. These tests revealed that each of the six chimpanzees selected grapes over similarly-sized carrot pieces ≥80% of the time. To determine that the chimpanzees would eat carrot pieces when no other food options were available, on a third day the chimpanzees were offered only carrot pieces and all the chimpanzees readily ate all the carrot pieces given to them. Thus, grapes were used throughout this study as the more-preferred food rewards and carrot pieces as the less-preferred food rewards.

## Procedure

Within each test session, the chimpanzees had to collect tokens (10 cm long lengths of 2.5 cm diameter white PVC pipe) from a single location. The chimpanzees could then exchange the tokens with researchers at one of two locations on the perimeter of their indoor exhibit; one where they could get a piece of carrot for each token exchanged (CLOSE location) and one where they could get a grape for each token exchanged (FAR location, Fig. 1). The distance from the location where the chimpanzees collected the tokens to the two exchange locations was different such that the chimpanzees had to travel farther to reach the FAR location than to reach the CLOSE location (Fig. 1 provides details of the specific distances). The entire study was subdivided into three phases (1, 2, and 3). To measure the chimpanzees' behavioral flexibility, we varied the location of the exchange locations across the phases (and therefore also the effort required to reach each location), as shown in Table 1 and Fig. 1. Each of the three phases was comprised of 30, thirty-minute test sessions (90 sessions in total = 45 h), with two test sessions run per week (on Tuesday and Thursday mornings from 11:30–12:00). The study lasted 15 months, and data were collected between January 2013 and April 2014.

At the beginning of every test session, a member of the animal care staff placed the tokens into two hoppers (plastic milk crates) hung side-by-side on the human side of a panel of mesh at the edge of the chimpanzees' enclosure (location A, shown in Fig. 1). The chimpanzees could easily obtain the tokens by reaching for them through the mesh with their fingers. To ensure that the tokens could not be monopolized by a single animal,

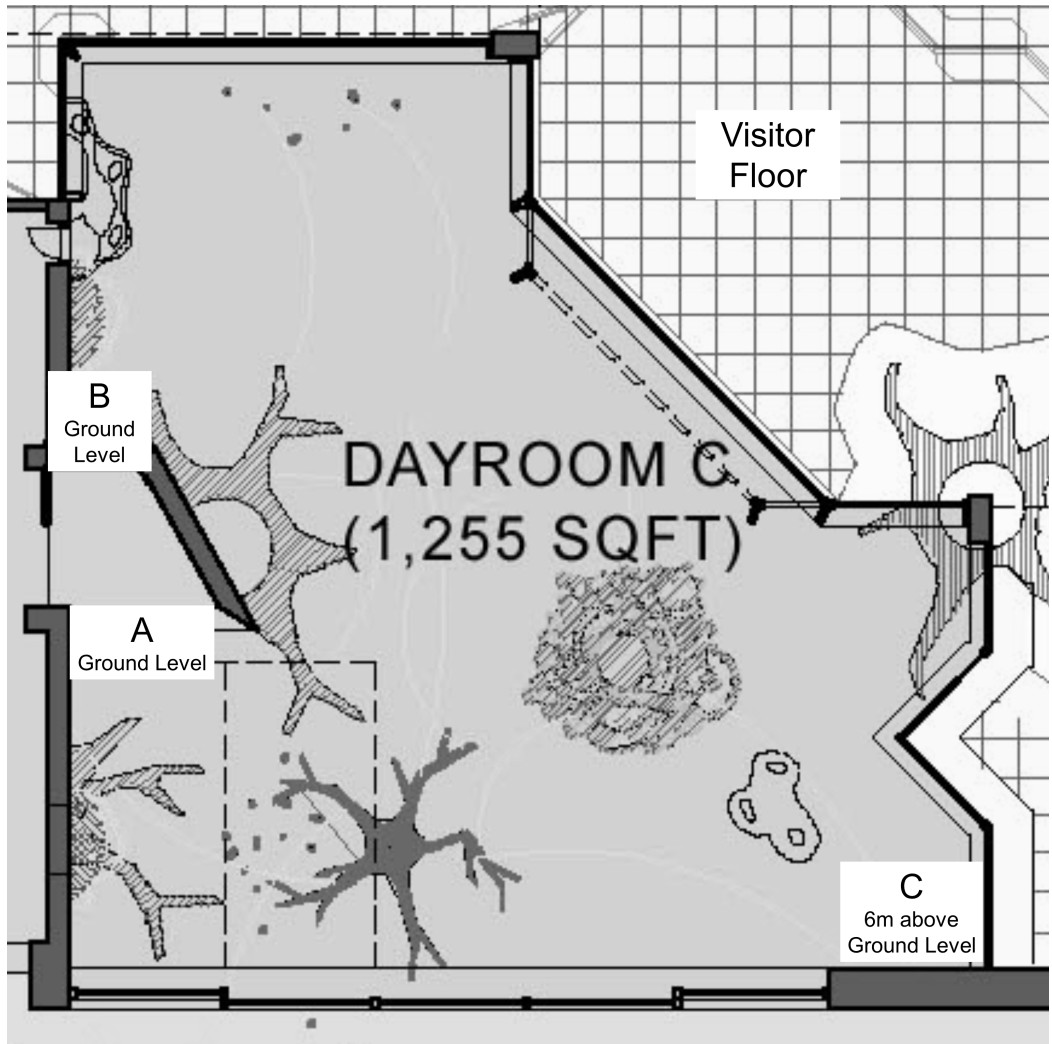

**Figure 1** **The chimpanzees' indoor enclosure and the locations where they could exchange tokens for food rewards during the study.** A plan of the chimpanzees' indoor enclosure at the Regenstein Center for African Apes, Lincoln Park Zoo also showing part of the visitor floor. Location A was where the chimpanzees could collect the PVC tokens from one of two hoppers hung on their cage mesh and, in phases 1 and 3, where they could exchange the tokens with a researcher to obtain less-preferred carrot pieces (CLOSE location, Table 1). Location B, which was a 10 m distance from location A following the shortest route, was where the chimpanzees could exchange tokens with the researchers for more-preferred grapes in phases 1 and 3 (FAR location) and less-preferred carrot pieces in phase 2 (when it was the CLOSE location, Table 1). For footage of the chimpanzees carrying tokens from location A to location B in phase 3, as viewed from the visitor floor, go to http://youtu.be/bl-byx754AI. Location C was where the chimpanzees could exchange tokens with the researchers for more-preferred grapes in phase 2 (FAR location, Table 1). To reach this location the chimpanzees had to walk from A for a minimum of 9 m and then climb 6 m to reach a mesh panel at the mezzanine level (15 m total); for footage of a chimpanzee traveling from location A to C while carrying tokens, go to: http://youtu.be/mC34z6vxXhk. Both locations A and B were at ground-level, while location C was 6 m above ground-level. Each exchange location was a discrete area, separated from other exchange locations by a minimum of 10 m, and the size of the mesh panels through which the chimpanzees could exchange tokens at these locations was 1.7 m × 2 m (location A), 1 m × 2 m (location B) and 1 m × 1 m (location C).

**Table 1 An overview of the three experimental phases.** The food rewards that were available at each of the three locations (shown in Fig. 1) in each of the three phases. Each phase was comprised of 30 thirty-minute sessions. The tokens, which the chimpanzees had to collect to exchange for the food rewards, were always available at location A. The distance from A to B was ≥10 m and the distance from A to C was ≥15 m, including a 6m climb to reach an elevated platform (Fig. 1).

|         | Location A              | Location B    | Location C |
|---------|-------------------------|---------------|------------|
| Phase 1 | Tokens + carrot pieces  | Grapes        |            |
| Phase 2 | Tokens                  | Carrot pieces | Grapes     |
| Phase 3 | Tokens + carrot pieces  | Grapes        |            |

the chimpanzees were provisioned with 150 tokens per session (25 tokens/chimpanzee) and could collect them from either of the two hoppers. The chimpanzees were able to collect as many tokens from the hoppers as they wished and exchange them with a researcher at either of the two exchange locations (Fig. 1). To successfully exchange a token, a chimpanzee had to completely push the token through the cage mesh in front of the researcher, but they did not hand it to the researcher directly (the zoo's safety policy prohibits such direct interactions between researchers and animals). In each session, a researcher stood at each of the two exchange locations (CLOSE and FAR) and gave the chimpanzees a food reward for each token that they exchanged. At each exchange location, a clear container filled with the food rewards was fully visible to any chimpanzees that approached the exchange location; at the CLOSE location the tub always contained pieces of carrots and at the FAR location the tub always contained grapes. The two researchers were the only two people with whom the chimpanzees could exchange the tokens or receive rewards from. To reduce the risk of researcher bias or chimpanzee preferences for certain researchers, the researchers switched locations every session. Across all phases, there was no difference in the number of tokens chimpanzees exchanged with either of the two researchers at either the CLOSE or FAR location (Wilcoxon Signed Rank test: CLOSE: $T = -.674$, $N = 6$, $P = 0.500$; FAR: $T = -0.943$, $N = 6$, $P = 0.345$). For each exchange that a chimpanzee completed, the researcher simply gave them the appropriate food reward and did not provide any other reinforcers (e.g., verbal praise) and nor did they encourage the chimpanzees to exchange tokens (e.g., by holding out their hand or by calling the chimpanzees over).

## Coding and analysis

All sessions were filmed using Sony Handycams (HDR-CX160; Sony, Tokyo, Japan). At each of the two exchange locations cameras on tripods were set to film the mesh panel through which the chimpanzees could exchange tokens (Fig. 1). This footage also captured any other chimpanzees within the vicinity of the exchange location (i.e., chimpanzee 'observers'). Throughout each session, each of the two researchers provided a running commentary that was recorded by the camera on the tripod at their location. Additionally, a third researcher with a handheld camcorder stood on the public floor and filmed the chimpanzees' activities (Fig. 1).

For each token that a chimpanzee exchanged, the researcher noted (1) the identity of the chimpanzee who completed the exchange, (2) the food reward that they were given (grape or piece of carrot), (3) whether the focal chimpanzee ate the food reward within 30 s, and, if not, whether another member of the group ate it within 30 s and that chimpanzee's identity. The researchers also noted which other chimpanzees, if any, observed each exchange. An 'observing' chimpanzee was defined as 'a chimpanzee that was within 1 m of the chimpanzee making the exchange and that was also oriented towards them as they completed the exchange with a researcher' (c.f., *Hopper et al., 2011*). The researcher also noted every time a chimpanzee exchanged an item from their exhibit that was not a token (e.g., a piece of bark from the mulch floor, or a stick from a shrub in their outside enclosure) and which chimpanzees observed these exchanges, if any. Exchanges of non-token items were never rewarded; the chimpanzees could only obtain food rewards from the researchers by exchanging one of the 150 provided PVC tokens. *Ad libitum* data were also collected on 'token transfers' that were observed by the researchers from the exchange locations. A token transfer was defined as 'one chimpanzee taking a token directly from the possession of another chimpanzee' (*sensu Hopper et al., 2011*) rather than collecting it from one of the two hoppers at location A or picking a token up from the floor. Footage of a token transfer event can be seen here: http://youtu.be/v0WYEcYn8Wo. All data were transcribed into Excel for analysis at which point the time stamp for each exchange (as gathered from the video camera recording) was also logged.

As two researchers simultaneously collected data, after the first two sessions, both researchers coded the video tape footage of both exchange locations (i.e., coding their own exchanges with the chimpanzees and those made by the second researcher). The two researchers had 100% concordance in their rating of each exchange that was completed and the identity of all observers. All further video footage was coded singly by LMK. Additionally, we calculated the reliability of our ratings by computing intra-class correlation coefficients (ICC) between the ratings completed by LMK and ratings completed by a researcher who was familiar with this chimpanzee group, but who had never seen these test sessions. This researcher blind coded two randomly-selected tapes of the chimpanzees' exchanges at the CLOSE location and two randomly-selected tapes of the chimpanzees' exchanges at the FAR location for each of the three experimental phases (12 tapes in total, 360 min). There was high inter-rater reliability for all measures; ICC (2,1) for the number of tokens exchanged by each chimpanzee within a session at a certain location $= 0.999$ ($P < 0.001$); ICC (2,1) for the number of non-token items exchanged by each chimpanzee within a session at a certain location $= 0.969$ ($P < 0.001$); and ICC (2,1) the number of observers that watched each chimpanzee's exchanges $= 0.876$ ($P < 0.001$). These inter-class correlation coefficients, along with all analyses reported below, were completed in IBM SPSS version 20 (IBM, New York, New York, USA), while all graphs were produced in R (*R Development Core Team, 2010*) using ggplot 2 (*Wickham, 2009*). In order to determine whether the chimpanzees' behavior (e.g., food scrounging) or their acquisition of the task was related to their rank, we collected assessments of the chimpanzees' rank from four researchers who work regularly with the chimpanzees

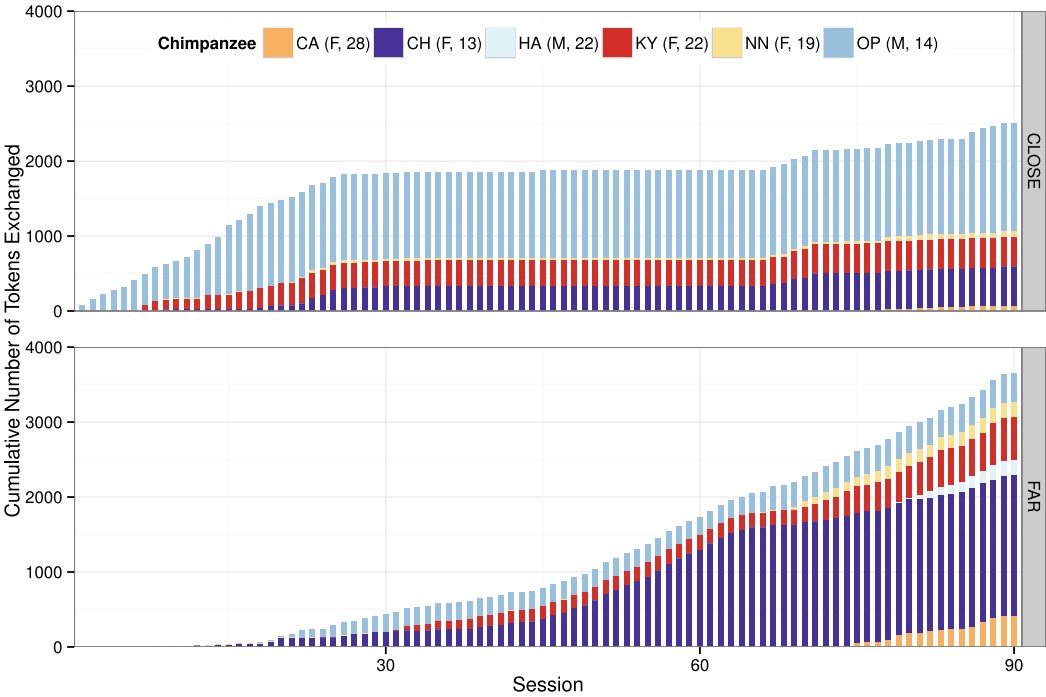

**Figure 2** **Cumulative total of token exchanges made by the chimpanzees at both the CLOSE (top) and FAR (bottom) exchange locations throughout the 90 sessions.** All six chimpanzees exchanged multiple tokens throughout this study although the alpha male (HA) only exchanged tokens in phase 3. Note too the plateau of exchanges in phase 2 (sessions 31–60) at the CLOSE location when a total of only 34 exchanges were made (by CH and OP, Table 2). During this phase, chimpanzees had to travel to obtain either reward, and so were required to carry their tokens 10 m to reach the CLOSE location, unlike in phase 1 (sessions 1–30) and phase 3 (sessions 61–90) in which no travel was required to reach the CLOSE location (chimpanzees could exchange their tokens where they collected them to obtain carrot pieces at location A)

(following *Kendal et al., 2015*). However, rather than using a categorical measure of rank (i.e., high, mid, low, c.f., *Kendal et al., 2015*), the chimpanzees' rank was rated on a linear scale from 1 (lowest ranked) to 6 (highest ranked).

## RESULTS

### Acquisition and adoption of the exchanging behavior

All chimpanzees in the group exchanged multiple tokens throughout the course of the study (Fig. 2). The first chimpanzee to exchange a token did so during the first session of phase 1 at the CLOSE location and the last chimpanzee to start participating first exchanged a token at the FAR location during the 78th session in phase 3 (locations A and B respectively, Fig. 1). There was no correlation between the chimpanzees' rank and the order in which they acquired this task (i.e., session in which they made their first exchange for a food reward food reward: Spearmans's rho: $r_s = -0.371, N = 6, P = 0.468$); however, there was a negative correlation between rank and number of tokens exchanged across all sessions (Spearmans's rho: $r_s = -1.00, N = 6, P < 0.001$). This pattern was reflected when considering their exchanges at the CLOSE (Spearmans's rho: $r_s = -0.886, N = 6,$

**Table 2 The number of tokens exchanged by each of the six chimpanzees at each exchange location in each phase.** The ID code for each chimpanzee also provides information about their sex (M or F), their age in years at the start of the study in January 2012, and the number presented outside the brackets is their average rank score where 1, least dominant and 6, most dominant.

| Chimpanzee | Phase 1 | | Phase 2 | | Phase 3 | |
|---|---|---|---|---|---|---|
| | CLOSE | FAR | CLOSE | FAR | CLOSE | FAR |
| CA (F, 28) 4 | 2 | 0 | 0 | 1 | 65 | 410 |
| CH (F, 13) 1 | 340 | 206 | 4 | 1,095 | 189 | 585 |
| HA (M, 22) 6 | 0 | 0 | 0 | 0 | 0 | 202 |
| KY (F, 22) 3 | 335 | 0 | 0 | 196 | 61 | 381 |
| NN (F, 19) 4 | 29 | 2 | 0 | 1 | 40 | 199 |
| OP (M, 14) 2 | 1135 | 227 | 30 | 0 | 270 | 149 |

$P = 0.019$) and FAR (Spearmans's rho: $r_s = -0.812, N = 6, P = 0.050$) exchange locations separately. In addition to exchanging the PVC tokens, some chimpanzees exchanged other items from within their enclosure (for which they were not rewarded). The rate of exchanging non-token items was relatively low (average = 2.8 exchanges/session, compared to an average 68.3 token exchanges/session) and the chimpanzees exchanged significantly more tokens than non-token items (Wilcoxon signed ranks test: $z = -2.20$, $N = 6, P = 0.028$).

## Selective exchanges for preferred rewards

Beyond simply evaluating whether the chimpanzees could learn how to trade tokens for food rewards, we were interested in assessing whether they showed a preference for exchanging tokens for their more-preferred food reward, grapes, despite having to travel farther to obtain them. (We also analyzed two other factors that might have influenced where the chimpanzees' exchanged the tokens—the number of visitors present and the chimpanzees' individual food preferences—but neither of these influenced the chimpanzees' behavior; see Supplemental Information 1 for these analyses.)

Over the 90 sessions, the chimpanzees made a total of 2,500 exchanges at the CLOSE locations and 3,654 at the FAR location (see Supplemental Information 2). In phase 1, the five chimpanzees that exchanged tokens exchanged more tokens at the CLOSE location for carrot pieces than at the FAR location for grapes (Wilcoxon signed ranks test: $z = -2.02$, $N = 5, P = 0.043$, Table 2). In phase 2, there was no significant difference in the location where the chimpanzees exchanged tokens (Wilcoxon signed ranks test: $z = 1.22, N = 5$, $P = 0.223$). This is most likely because only three chimpanzees exchanged >10 tokens in this phase, but the two females that exchanged the most tokens in this phase did so at the FAR location (Table 2). In phase 3, when the exchange locations were the same as in phase 1 (Table 1), and all six chimpanzees exchanged tokens, the chimpanzees exchanged more tokens at the FAR location for grapes than at the CLOSE location for carrot pieces (Wilcoxon signed ranks test: $z = 1.99, N = 6, P = 0.046$, see Table 2 and the Supplemental Information 2).

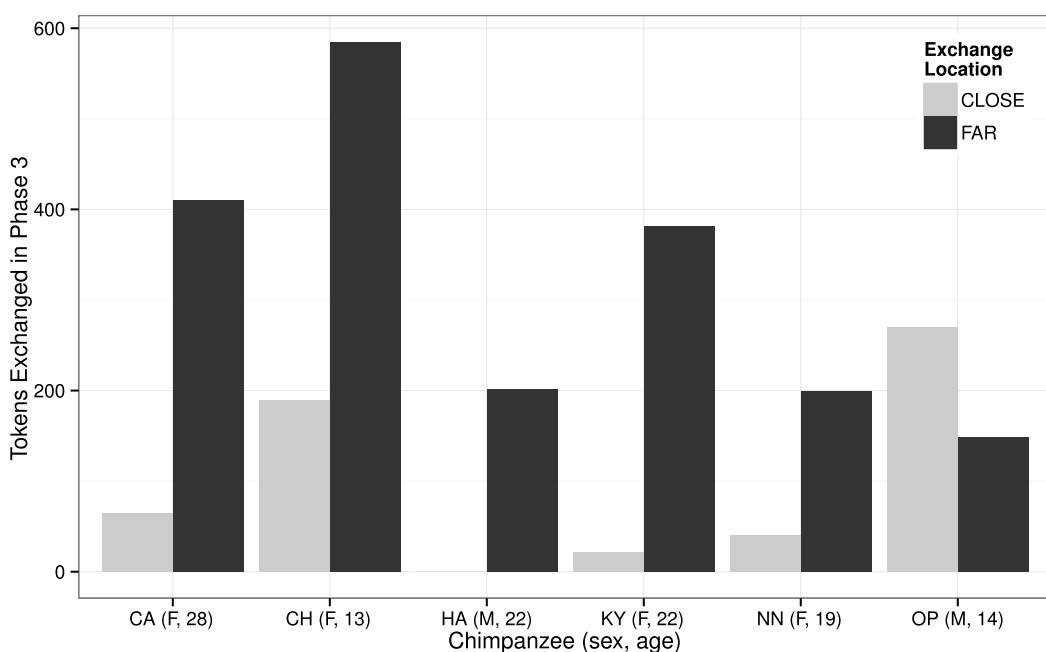

**Figure 3 The total number of tokens that each of the chimpanzees exchanged at the CLOSE and FAR locations in phase 3.** The ID code for each chimpanzee also provides information about their sex (M or F) and their age in years at the start of the study in January 2012. Note that male HA only exchanged tokens at the FAR location and never at the CLOSE location.

The difference in the chimpanzees' responses between phase 1 and 3 is most likely due to the chimpanzees' acquisition of the task; in phase 3, when the chimpanzees had greater experience with the protocol and more chimpanzees participated in the study, they were more likely to exchange their tokens for the more-preferred grapes (Fig. 3; Supplemental Information 2). The chimpanzees' behavior in phase 3 (Fig. 3) suggests that they preferred to travel 10 m to obtain grapes than to directly exchange tokens for carrot pieces, which required no travel. We propose that their behavior in phase 3 is a more accurate reflection of their choices because, by this phase, all six chimpanzees had exchanged tokens for food rewards, all had greater exposure to the task, and all had discovered the FAR location. Supporting this, as their exposure to the task in phase 1 increased, the number of exchanges that the chimpanzees made at the FAR location increased over time. Specifically, the number of tokens exchanged at the FAR location was positively correlated with session number (Spearmans's rho: $r_s = 0.858$, $N = 30$, $P < 0.001$) and conversely there was a negative correlation between number of tokens exchanged at the CLOSE location and session number (Spearmans's rho: $r_s = -0.469$, $N = 30$, $P = 0.009$).

## Switching strategies and spatial discounting

Aside from one male, all chimpanzees exchanged tokens at both the CLOSE and FAR locations (Fig. 2). Thus, even after discovering one successful method, the chimpanzees explored alternative options (Table 2). Indeed, these five chimpanzees also exchanged tokens at both locations within single sessions, but there was no correlation between the chimpanzees' rank and the average number of times they switched between the

**Table 3 The number of exchanges that each subject observed at each location in each phase.** The first number is the total number of exchanges observed by a chimpanzee and the number shown in brackets is the number of exchanges observed by a chimpanzee before they made an exchange themselves at that location and '-' indicates that they never made an exchange at that location within that phase. The ID code for each chimpanzee also provides information about their sex (M or F), their age in years at the start of the study in January 2012, and the number presented outside the brackets is their average rank score where 1, least dominant and 6, most dominant.

| Chimpanzee | Phase 1 | | Phase 2 | | Phase 3 | |
|---|---|---|---|---|---|---|
| | CLOSE | FAR | CLOSE | FAR | CLOSE | FAR |
| CA (F, 28) 4 | 120 (21) | 10 (-) | 0 (-) | 30 (28) | 55 (3) | 312 (32) |
| CH (F, 13) 1 | 256 (49) | 7 (0) | 0 (0) | 158 (0) | 43 (0) | 314 (0) |
| HA (M, 22) 6 | 20 (-) | 19 (-) | 2 (-) | 9 (-) | 6 (-) | 237 (125) |
| KY (F, 22) 3 | 238 (78) | 6 (-) | 0 (-) | 96 (7) | 55 (0) | 287 (21) |
| NN (F, 19) 4 | 317 (19) | 9 (0) | 0 (-) | 157 (69) | 32 (7) | 482 (0) |
| OP (M, 14) 2 | 155 (0) | 0 (0) | 0 (0) | 8 (-) | 26 (2) | 141 (0) |

two exchange locations within in a single session (Spearmans's rho: $r_s = 0.200$, $N = 5$, $P = 0.747$).

In phase 1, three chimpanzees, who had previously received rewards for exchanging tokens at the CLOSE location, went on to exchange tokens at the FAR location. None of these three chimpanzees had been recorded to observe any chimpanzee exchange a token at the FAR location before they themselves first exchanged a token at that location (Table 3). Furthermore, despite introducing novel exchanging locations in phase 2, four chimpanzees exchanged tokens at the new FAR location (Table 2), and, aside from the first female to discover this new location, all did so after observing other chimpanzees at that location successfully exchange tokens for grapes (Table 3). In phase 3, when the two exchange locations mirrored those in phase 1, three of the chimpanzees exchanged tokens with researchers at the FAR location for the first time, having not done so in phase 1, and all did so after they had observed multiple successful exchanges at this location by other chimpanzees in their group (Table 3). In phase 3, five of the chimpanzees exchanged at both the CLOSE and FAR locations, and continued to exchange at the CLOSE location even after discovering the FAR one (Table 2).

In phase 2, the reward locations were different to those in phases 1 and 3, and in this phase the chimpanzees were required to travel to obtain both reward types (Fig. 1). The chimpanzees varied in their willingness to travel 15 m, compared to 10 m, to obtain the more-preferred grapes and 10 m, compared to 0 m, to get carrot pieces. Considering their exchanges for grapes, most of the chimpanzees made more exchanges at the FAR location in phase 3 compared to phase 1 (Table 2), most likely because they were more familiar with the task, as discussed above. In phase 2, when the effort to obtain grapes was increased, the majority of chimpanzees made fewer exchanges at the FAR location (Table 2), compared to in phases 1 and 3. One 13-year old female (CH), however, exchanged more tokens in phase 2 at the FAR location (when she had to travel 15 m) than in either phase 1 or 3 (when she only had to travel 10 m to obtain grapes).

## Strategies for reducing the effort required to obtain rewards

During the study, the chimpanzees adopted three key strategies that potentially reduced the effort required to obtain the rewards. One was to take tokens from other chimpanzees who had already retrieved them from the token hopper and carried them to an exchange location (a 'token transfer'); a second was to eat the food reward after an exchange had been completed by another individual (a 'food scrounging' event); and a third was to carry more than one token from location A to the exchange locations. Across all three phases, there were 172 token transfer events, and although the chimpanzees exchanged the token they obtained in 84.4% of these cases, these events represented only 2.4% of the 6,154 total exchanges made by chimpanzees throughout the three phases. The rate of reward scrounging was also very low: 317 food rewards were scrounged in total during the 90 sessions.

Considering the chimpanzees' scrounging behavior, there was no difference in the number of tokens chimpanzees took from their group mates across the three phases (Friedman's test: $X^2(2) = 3.36$, $P = 0.186$). In phase 3, however, chimpanzees were more likely to take tokens from their group mates and exchange them within 30 s for rewards at the FAR location compared to at the CLOSE location (Wilcoxon signed ranks test: $z = 2.03$, $N = 6$, $P = 0.042$). However, there was no difference in the number of tokens that chimpanzees took from their group mates at the two exchange locations in either phase 1 (Wilcoxon signed ranks test: $z = -0.82$, $N = 5$, $P = 0.414$) or phase 2 (Wilcoxon signed ranks test: $z = 1.00$, $N = 3$, $P = 0.317$). Reflecting their token transfer behavior, there was no difference in the number of food rewards that the chimpanzees scrounged from their group mates across the three phases (Friedman's test: $X^2(2) = 2.33$, $P = 0.311$). In phase 1, when the chimpanzees made more exchanges for carrots than grapes, the chimpanzees scrounged more carrots from their group mates at the CLOSE location than grapes at the FAR location (Wilcoxon signed ranks test: $z = -1.00$, $N = 6$, $P = 0.046$). In phases 2 and 3, when the chimpanzees exchanged more tokens for grapes, there was a trend (although not significant) for the chimpanzees to scrounge more grapes at the FAR location than for carrots at the CLOSE location (Wilcoxon signed ranks test: phase 2, $z = 1.83$, $N = 6$, $P = 0.068$; phase 3, $z = 1.75$, $N = 6$, $P = 0.080$).

There was no correlation between the chimpanzees' rank and the number of tokens (Spearmans's rho: $r_s = -0.086$, $N = 6$, $P = 0.872$) or food rewards (Spearmans's rho: $r_s = 0.429$, $N = 6$, $P = 0.397$) that they scrounged. However, considering the two types of scrounging behavior collectively (food scrounging plus token transfers), there was a positive correlation between the chimpanzees' rank and the proportion of scrounging events that were food scrounging events (Spearman's rho: $r_s = 0.899$, $N = 6$, $P = 0.015$). Specifically, higher ranked individuals were more likely to scrounge food rewards while lower ranked individuals were more likely to take tokens from their group mates.

In phases 1 and 3, the upshot of the chimpanzees sometimes carrying more than one token with them from location A to location B was that the average latency between each exchange a chimpanzee made at the FAR location was not significantly different than the latency between exchanges made at the CLOSE location, even though travel between exchanges was not required at the CLOSE location (Wilcoxon signed rank test: phase 1,

$z = 0.00$, $N = 3$, $P = 1.00$; phase 3, $z = 1.75$, $N = 5$, $P = 0.08$). In phase 2, only one chimpanzee exchanged tokens at both the CLOSE and FAR locations.

## DISCUSSION

The purpose of this study was to document the acquisition of a novel foraging paradigm involving token exchange by a group of zoo-housed chimpanzees. Beyond this, we were interested in the chimpanzees' flexibility to discover and exploit novel exchange locations within a social environment, and also to document how effort impacted their foraging choices. As predicted, all six chimpanzees learned how to exchange tokens in order to obtain food rewards from researchers, but even within this small sample of six chimpanzees there was individual variation among the chimpanzees in the time it took for them to do so and their overall level of participation in the study. The chimpanzees exchanged significantly more tokens than non-token items (e.g., bark chips and twigs), which suggests that they learned the contingency of exchanging tokens specifically, rather than just general exchanging behavior. Once all the chimpanzees were proficient at exchanging tokens, they chose to travel farther to exchange tokens for a preferred food (grapes), but sometimes adopted strategies to reduce the effort per exchange, by scrounging tokens and food rewards from others and by carrying more than one token at a time. However, when the distances to reach both rewards were increased in phase 2 (sessions 31–60), the chimpanzees were less likely to travel to obtain either reward. Finally, aside from the alpha male, in phase 3 (sessions 61–90) all the chimpanzees exchanged tokens at both the CLOSE and FAR locations demonstrating their ability to explore the available options, even after they had previously learned one solution. It was this flexibility that allowed the chimpanzees to discover their preferred exchange location where they could obtain grapes (the FAR location).

The chimpanzees learned the exchanging paradigm very quickly. Five of the six chimpanzees made their first exchange within the first seven sessions, and the first chimpanzee did so within the first four minutes of the first session. These five chimpanzees all completed their first exchange at the CLOSE location, which was also where they collected the tokens. Although it is possible that these individuals had previously learned exchange contingencies from interactions with animal care staff, it is important to note that they were not specifically trained to do so, nor did they have any experience receiving food rewards from the researchers prior to this study. Intriguingly, despite the relatively quick acquisition of the required exchanging behavior by the chimpanzees, the dominant male did not make his first exchange until the 78th session. His first exchange was made after he had been exposed to the experimental paradigm for 13 months and had seen his group mates rewarded for 181 exchanges (including observations of exchanges at both locations in each of the three phases), which suggests that his lack of participation cannot be explained by a lack of opportunities to observe other individuals performing the task. It is notable that this same male was also the last to start using tools in an earlier study that investigated the groups' acquisition of tool use when presented with a novel artificial termite mound in a social setting (*Lonsdorf et al., 2009*; *Hopper et al., 2015*).

In the third phase, when all six chimpanzees participated in the study, they showed a preference for exchanging tokens for their more-preferred rewards (grapes), even though they could obtain carrot pieces at the same location where they collected the tokens. The behavior of the chimpanzees both reveals their foraging preferences but also highlights their flexibility; five of the six chimpanzees exchanged at both the CLOSE and FAR locations, often within single sessions. Previous studies have suggested that chimpanzees may be conservative and unable to adopt new strategies if they already know one that is rewarding (*Hrubesch, Preuschoft & van Schaik, 2009*), a factor that may be related to chimpanzees' lack of cumulative culture (*Dean et al., 2012*). However, a more recent study revealed that chimpanzees were able switch away from a previously-learned location for exchanging tokens when a new one garnered more rewards (*Van Leeuwen et al., 2013*). Our results reflect this: five of the six chimpanzees exchanged at the FAR location for grapes, even though it required farther travel to reach and they had previously been rewarded for exchanging tokens at the CLOSE location.

Like tests of temporal discounting, in which individuals are asked to choose between a small reward now or a larger reward later (*Evans et al., 2012*), tests of spatial discounting assess whether individuals prefer to travel farther to obtain a more desirable reward, and how far they are willing to travel for that reward (*Stevens et al., 2005*; *Lihoreau, Chitkka & Raine, 2011*; *Perrings & Hannon, 2001*; *Kralik & Sampson, 2012*). For example, when given the choice to walk less far to a smaller reward or farther for a bigger reward, tamarins (*Saguinus oedipus*) consistently selected the larger reward regardless of distance; however, marmosets (*Callithrix jacchus*) were less likely to choose the larger reward over the smaller one as the distance between the two options increased (*Stevens et al., 2005*). The responses of the chimpanzees in phases 2 and 3 suggest that they were willing to travel farther for better, but only up to a point; however, other factors may also have influenced their behavior. For example, while we controlled the distance that the chimpanzees had to walk, we could not control for the time it took for different individuals to walk from location A to B (as in tests of temporal discounting) and nor did we control the number of tokens that they carried per journey. Indeed, the chimpanzees sometimes carried multiple tokens, which had the end result of making chimpanzees' inter-exchange interval at the CLOSE and FAR locations not significantly different. Thus, the chimpanzees preferred to exchange tokens for grapes and developed strategies to reduce the effort required to obtain them, again highlighting their flexible problem solving skills.

Despite having never been in close range (within 1 m) of another chimpanzee exchanging tokens at the FAR location, and having successfully exchanged multiple tokens themselves for pieces of carrot at the CLOSE location, three chimpanzees explored and found the FAR location in phase 1 and exchanged tokens there. Due to the design of our study, we cannot know how each of these three chimpanzees discovered this novel exchanging location, whether by individual trial-and-error learning or via social means. Although it is likely that all three chimpanzees independently discovered this new location, especially as the test environment was their familiar exhibit, as soon as the first chimpanzee exchanged a token at the FAR location, we cannot rule out the possibility that the other

chimpanzees to exchange there were influenced by the behavior of the first. Even if these chimpanzees were never recorded as being within 1 m of the first female when she exchanged a token at the FAR location before they themselves did, simply seeing her carrying tokens to the FAR location, in combination with their personally-acquired knowledge that tokens could be exchanged with researchers to obtain foods, might have been sufficient for them to learn this new option. This 'low-level' social learning mechanism (i.e., local enhancement) has also been shown to influence the learning and decision-making of a number of species when foraging (e.g., *Mikolasch, Kotrschal & Schloegl, 2012*; *Takahashi, Masuda & Yamashita, 2013*; *Avarguès-Weber & Chittka, 2014*; *Webster & Laland, 2013*) and thus is a parsimonious explanation for the chimpanzees' learning here.

Interestingly, the first female to exchange a token at the FAR location in phase 1 was the lowest-ranked member of the group and also the first to discover the novel FAR location in phase 2. It is possible that this 13-year old female's low rank or age may explain her exploratory behavior, as has been reported for wild chimpanzees (*Reader & Laland, 2001*). Rather than exhibiting behavioral inhibition in the presence of dominants and ending participation altogether, as reported previously for primates (e.g., *Macaca mulatta*, *Drea & Wallen, 1999*; *S. boliviensis*, *Hopper et al., 2013*; *P. troglodytes*, *Cronin et al., 2014*), this female continued to exchange tokens, but at alternative locations. Beyond this specific female, and contrary to our predictions based on the findings of previous research (e.g., *Bonnie et al., 2007*; *Addessi et al., 2011*; *Hopper et al., 2011*), in this group of six chimpanzees there was a negative correlation between rank and total number of exchanges made (i.e., low-ranked individuals made more exchanges than high-ranked individuals).

Even though the chimpanzees scrounged food and tokens from each other, the rate of scrounging that we observed was considerably lower than that reported for other token exchange studies run with groups of chimpanzees (e.g., *Hopper et al., 2011*). Furthermore, we found no correlation between rank and the total number of tokens and food rewards scrounged. However, we did find that dominants were more likely to take food rewards than tokens. We propose that the low rate of scrounging in our study, and the negative correlation between rank and total tokens exchanged, is because, unlike the paradigm used by *Hopper et al. (2011)*, the chimpanzees had two spatially distant locations within their enclosure where they could exchange tokens for food items. Additionally, we provisioned our chimpanzees with a greater number of tokens to exchange compared to previous studies, which also likely reduced within-group competition and therefore scrounging rates (e.g., this study provided = 25 tokens/subject while, for example, (*Bonnie et al., 2007*) provided 1.7 tokens/subject). Our experimental design, in combination with the small number of chimpanzees that this group is comprised of, may help to explain why, unlike in previous studies (e.g., *Bonnie et al., 2007*; *Addessi et al., 2011*), lower-ranking chimpanzees were able to exchange more tokens than higher-ranking individuals (they could avoid dominants by exchanging at alternative locations) and levels of scrounging were relatively low (tokens were not easily monopolized).

Ultimately, this study demonstrated that this group of six chimpanzees showed flexible foraging strategies and the ability to explore their environment in order to find their most

desired rewards. Following previous tests of foraging and decision making with captive animals (e.g., *Stevens et al., 2005*; *Lihoreau, Chitkka & Raine, 2011*; *Reilly et al., 2012*), beyond simply testing their ability to discover reward locations, we also assessed whether the chimpanzees would still attempt to obtain their more-preferred rewards when more effort (i.e., the distance required to reach them) was required. The chimpanzees, to a limit, were willing to travel farther to obtain the more-preferred grapes but future studies should test whether chimpanzees would be willing to exert more effort for more-preferred rewards if the type of effort was different (for example, time to process the food or increased competition to negotiate). Unlike in previous studies with chimpanzees (e.g., *Hrubesch, Preuschoft & van Schaik, 2009*; *Hopper et al., 2011*), the chimpanzees were not conservative, but showed the ability to transition between different solutions in order to maximize their rewards (see also *Manrique, Völter & Call, 2013*; *Van Leeuwen et al., 2013*; *Yamamoto, Humle & Tanaka, 2013*).

In captivity, primates are provisioned with food by human caretakers and so have reduced choice over the options available to them and, unlike wild primates that spend a large proportion of their waking hours foraging for food, captive primates are typically fed following a regular schedule (*Bloomsmith & Lambeth, 1995*). Therefore, this study was not only of academic interest but also created novel foraging enrichment for this group of chimpanzees and encouraged them to explore their environment to find the best possible foods. Indeed, a recent review of tool-use innovation by wild primates suggested that chimpanzees are more likely to innovate when the local ecology supports it (i.e., invention due to opportunity, not necessity, *Koops, Visalberghi & van Schaik, 2014*) and, in modest terms, we provided our zoo-housed group of chimpanzees with opportunities for innovation and discovery. This study also highlighted the individual differences among the chimpanzees such that some were quick to discover new options while others took longer to exchange tokens at all, but with such a small sample, further studies are required to determine how generalizable our findings are. These individual differences are likely a combination of the chimpanzees' rank (*Reader & Laland, 2001*) and their personality characteristics (*Freeman & Gosling, 2010*), which have been shown to correlate with chimpanzee problem-solving abilities in other foraging studies (*Massen et al., 2013*; *Hopper et al., 2014b*).

## ACKNOWLEDGEMENTS

Thanks to Katherine Cronin for her constructive feedback on earlier drafts of this article and for the comments given to us by three reviewers and the editor, Jennifer Vonk. Additionally, we thank Maureen Leahy, Michael Brown-Palsgrove, and the animal care staff in the Regenstein Center for African Apes at Lincoln Park Zoo, for making this research possible and for providing the highest level of care for the animals housed there. We are also grateful to Marisa Shender for assisting with data collection and to numerous Fisher Center interns for collecting video footage during test sessions. We also thank Sara Skiba, who blind coded footage for us, and Andrew Steets for his expert assistance with R and ggplot2.

### Funding

This study was funded by the Leo S. Guthman Fund, which also provides salary support for Lydia M. Hopper and Laura M. Kurtycz. The funders had no role in study design, data collection and analysis, decision to publish, or preparation of the manuscript.

### Grant Disclosures

The following grant information was disclosed by the authors:
Leo S. Guthman Fund.

### Competing Interests

When this study was run, Lydia M. Hopper, Laura M. Kurtycz and Stephen R. Ross were all employees of Lincoln Park Zoo. Kristin E. Bonnie was an (unpaid) Adjunct Scientist at Lincoln Park Zoo.

### Author Contributions

- Lydia M. Hopper conceived and designed the experiments, performed the experiments, analyzed the data, wrote the paper, prepared figures and/or tables, reviewed drafts of the paper.
- Laura M. Kurtycz performed the experiments, analyzed the data, wrote the paper, reviewed drafts of the paper.
- Stephen R. Ross conceived and designed the experiments, wrote the paper, reviewed drafts of the paper.
- Kristin E. Bonnie conceived and designed the experiments, analyzed the data, wrote the paper, reviewed drafts of the paper.

### Animal Ethics

The following information was supplied relating to ethical approvals (i.e., approving body and any reference numbers):

This study was approved by the Lincoln Park Zoo Research Committee, which is the governing body for all animal research at the institution. This research adhered to legal requirements in the United States of America and to the American Society of Primatologists' Principles for the Ethical Treatment of Nonhuman Primates.

### Supplemental Information

Supplemental information for this article can be found online at http://dx.doi.org/10.7717/peerj.833#supplemental-information.

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
