# Peer review of "Captive chimpanzee foraging in a social setting: a test of problem solving, flexibility, and spatial discounting"

_PeerJ, doi:10.7717/peerj.833_

## Round 0.1 · original submission · Minor Revisions

Three expert reviewers have now provided very helpful and thoughtful reviews of your work. I would like you to revise the manuscript taking their astute comments into account. I agree with the assessment and recommendations of the reviewers and add only a few comments of my own.
You do an excellent job of reviewing relevant, recent material in a concise manner. The ability to examine decision-making in primate groups has not been studied extensively and is an exciting area for future research. I do agree with Reviewer 2 that the Introduction should be reorganized to flow from general to specific and finally to the current study goals.

The abstract is a little lengthy and could be made more concise. For one, I would remove the sentence “Beyond innovators, we propose most chimpanzees discovered the alternative exchange locations from observing their group mates” as I am not sure it makes sense as written. I would reduce references to innovation throughout as I don’t think this study tests for innovation.

A study by Menzel that demonstrated that chimpanzees can remember the location of hidden foods and bypass less preferred foods, or organize a route to preferentially reach preferred fruits, first should be cited. It is important to integrate findings from lab studies into discussion on wild primates given that your study is conducted in a much more restrictive captive environment.

You manipulate distance travelled but no other aspect of “effort”. It is possible that animals weigh more heavily effort required to obtain or extract rewards, or risk of competition etc., which might be confounded with distance in your study. At least discuss as a limitation in the current research.

If you are interested in how chimpanzees modify their foraging decisions based on competition it would make sense to manipulate competition at the different sites, at least by releasing different numbers of chimpanzees into the habitat. If unable to do this, I would be cautious in the discussion of competition as a factor.

Did chimpanzees ever exchange multiple tokens for rewards on the same trip? That is, if they carried four tokens to the FAR location, the cost per grape is minimized compared to carrying one token at a time. Can you comment on this? If it occurred at reasonable rates you may have to analyze the data by trips taken rather than by exchanges per se as the cost factor is mitigated once the chimp has already travelled. In general, the results could be presented in a more systematic, more statistically driven manner, as the reviewers also suggest, rather than reading as descriptive and exploratory. You fluctuate between group level and individual level analyses. Perhaps a multi-level model approach would be useful including dominance and/or age as a factor.

In addition to reducing competition, you might consider that chimpanzees continued to trade tokens for carrots to reflect a desire for a variable diet, rather than one that contains exclusively preferred foods.

Should “exchange” be written in the past tense on line 99?
Chimpanzees should be plural on line 117
On line 151, should this not be a reference to Hopper et al. 2014b, rather than a?
Should line 174 be “predictions” rather than predications?
Should line 822 be “shortest” route?

·

Basic reporting

The article ‘Captive chimpanzee foraging in a social setting: a test of problem solving, flexibility, and spatial discounting’ is well written and in a clear English. The text contains sufficient introduction and background to demonstrate how the work fits into the broader field and the relevant literature is cited appropriately.

Figures and tables:
Figure 1 is not clear, please reformulate the legend explaining first all the location for phase 1 and 3 and then all the locations for phase 2… Or change for 2 more explicit figures with tokens, carrots and grapes logo instead of A, B, C . Reading the end of the section I see that table 1 summarize it all in a clearer way, cite this table earlier in text when you cite Fig.1 (line_ 281).

In Figure 2, the different grey colours are not very visible… Better to change for different colours or different black and white pattern.

In table 3, to have a clearer legend: Add: ‘the first number is the total number of exchanges observed by a chimpanzee and’ before ‘The number shown in brackets is the number of exchanges observed by a chimpanzee before they made an exchange themselves at that location and ‘-‘ indicates that they never made an exchange at that location within that phase. ‘

Add a number (1 to 6) represent the rank next to sex and age for your individuals in your tables.

Experimental design

The exchanges of tokens are done with researchers, this can have an influence on some findings like which chimp exchanges tokens and where chimps go depending which researcher is where? It would have been better with automatic dispenser… Please check if they are any biases of location due to researcher identity at the different locations.

Location of B and C are kind of opposite. It would be good for a next study to test a design where to go to the farther location you pass the closer one to see if they can restrain themselves to exchange at the close location to reach the preferred food at the farther one… I think a set-up link this would mimic better the natural environment where you have in a forest many trees of the less preferred food type and some rare trees of the preferred type.

A potential bias of this study is that during phase 1 & 3 the carrots are seen when taking the tokens as in same location whereas in phase 2 no food is at the location of the tokens. Thus the findings that more carrots are eaten in trial 1 might have nothing to do with spatial discounting but only with a temporal one, as the chimps can get carrots directly now after taking a token whereas for the grape they will only get it later. In this setup carrots next to the tokens might be their preferred strategy, thus monopolized by dominants explaining part of your results of low-ranking eating more of the ‘preferred’ grapes when at equal distance than carrots but maybe not they are not preferred anymore when further away… Add a sentence or two about time and spatial discounting in discussion linking them together and discussing more your results in this view.

Validity of the findings

A potential bias of this study is that during phase 1 & 3 the carrots are seen when taking the tokens as in same location whereas in phase 2 no food is at the location of the tokens. Thus the findings that more carrots are eaten in trial 1 might have nothing to do with spatial discounting but only with a temporal one, as the chimps can get carrots directly now after taking a token whereas for the grape they will only get it later. In this setup carrots next to the tokens might be their preferred strategy, thus monopolized by dominants explaining part of your results of low-ranking eating more of the ‘preferred’ grapes when at equal distance than carrots but maybe not they are not preferred anymore when further away… Add a part in discussion about time and spatial discounting in discussion linking them together and discussing more your results in this view.

I also think that your results about the social settings of your experiments should be interpreted with care. You need to make more links to natural groups composition where 6 would be only a subgroup foraging on its own… And explain the restrictions of your findings in a captive setup to explain a natural group dynamic to forage.

Additional comments

The authors wrote an interesting paper that answered well their various but complementary aims: to test the chimpanzees’ ability to learn a novel foraging task; to determine whether the chimpanzees could switch between different foraging patches as the reward values available at each changed; to assess how the chimpanzees responded to the influences of their social environment; and to record whether the chimpanzees were willing to travel further to obtain a more-preferred reward and whether this preference remained after the distances for both reward values increased.

Reviewer 2 ·

Basic reporting

This article describes a study aiming to assess: (i) whether naïve chimpanzees acquire token exchange without explicit training, (ii) if they exchange tokens according to their food preferences, (iii) how their rank affects exchanging behavior, and (iv) whether scrounging behavior and token transfer between subjects play a role. Six chimpanzees were tested in 3 phases, each involving thirty 30-min sessions, in which they could exchange tokens with two experimenter positioned at two different locations – a close location and a far location. In all phases, the experimenter at the close location would provide one piece of low-preferred food for each token exchanged, whereas the experimenter at the far location would provide one piece of high-preferred food for the same behavior. Overall, chimpanzees learned to exchange tokens with the experimenters and, although exploited both exchange locations, exchanged more tokens for high-preferred food. Low-ranked individuals exchanged more tokens than high-ranked ones, whereas high-ranked individuals scrounged more food than low-ranked ones.

- The paper is generally well written, although the introduction needs to be better structured from line 60 onwards. Specifically, the current study should be introduced after reviewing the relevant literature, rather than beginning to present this study, then reporting a description of previous studies, and finally going back to the current study and its aims. Moreover, some of the aims lack the corresponding predictions, namely the second aim (ll 160-173), and the last aim (ll 201-206)

- In the Introduction and elsewhere in the paper, I suggest to give less weight to the spatial discounting argument, since the study does not seem specifically tailored to investigate this issue. Although the study design potentially implies that the farthest option might be discounted, in my opinion this aspect should only be discussed in the Discussion section (where at the moment it is apparently not mentioned), without presenting it among the aims of the study. When mentioning spatial discounting, you may want to cite also the following article:
Kralik JD, Sampson WW (2012) A fruit in hand is worth many more in the bush: steep spatial discounting by free-ranging rhesus macaques (Macaca mulatta). Behav Processes 89(3):197-202.

- Introduction, ll 67-69: “Additionally, we were also able to change the rewards that were available over time…shift their choices in responses to the fluctuating resources”: this sentence is not clear, since the rewards available were always the same, although at different locations

Experimental design

- ll 240-242: according to the PeerJ policy, research on non-human primates is subject to specific guidelines from the Weatherall (2006) report (The Use of Non-Human Primates in Research), please specify whether your study adhered to its requirements (I guess so)

- ll 231-232: how did the presence of zoo visitors potentially impact on chimpanzees’ behavior? From Figure 1 it seems that zoo visitors could approach location C, was this so?

- ll 247-249: how was the interaction between the tested individual and the other group members avoided?

Validity of the findings

- ll 362-363: please modify as follows: “session in which they made their first exchange for a food reward”

- ll 363-365 and elsewhere in the Results section: why did you carry out Spearman’s correlation for some series of data and Pearson’s correlation for others? Please either clarify (especially with respect to whether the assumptions of parametric statistics were met) or always use Spearman’s correlation

- ll 371-372: is this the cumulative frequency rather than the average number of tokens exchanged/session? In the latter case, how could you obtain this large value since you provided 150 tokens per session? The same comment applies to the label of the y-axis in Figure 2

- ll 427-429: was a statistical comparison performed?

- l 435: “low-ranking female”: could you please specify which rank position she had?

- l 442: “only two chimpanzees exchange a total of 34 times in this phase [2]…”.: from Figure 2 it seems that in this phase three subjects actually exchanged at the close location, please clarify

- ll 457-458, 460-461, 465-466: please provide N for the Friedman’s tests

- ll 470-472: please provide the statistical analyses rather than only a qualitative description of these two results

- ll 483 and 487: please clarify why N = 53 or 83 rather than N = 90

- l 507: from ll 432-437 it seems that the subordinate female was just one, whereas here it seems that you make a more general, but unsupported, claim

- ll 528-529: rather than “lack of interest in the food rewards” I would say “lack of opportunities to observe other individuals performing the task”

- ll 542-543: “…it is likely that the second two of the three chimpanzees…”: this sentence is awkward, please clarify

- ll 544-553: please further tone down this statement, the more parsimonious explanations here is that all the three chimpanzees acquired the behavior by themselves

- ll 597-598: I could not find this comparison in the Results section

- l 603: “exchange a token” (typo)

- l 629: a full stop is missing at the end of the sentence

- please provide at the end of the Discussion a sentence in which you recognize that the small sample size may be a limit of the present study

Reviewer 3 ·

Basic reporting

The study adheres to the PeerJ policies, is well-written, well-embedded in the literature and the submission contains a complete publication package. No further comments.

Experimental design

The study's strength is its simplicity, quite nice. The design is adequately matched to the questions under study and the explanation of both the design and the resulting behavioral patterns are clear. The study also addresses a new question for chimpanzees, building nicely on previous research. In my view, all is clearly defined, quite relevant, and the study has been executed rigorously. The methods used are very good in many respects, yet the sample size is the obvious pitfall, see comments in next area.

Validity of the findings

Where the study excels with a nice and clear design, the sample size is unfortunately very low. Especially when conclusions are (aimed to be) drawn in light of further distinguishing features like dominance (and potentially individual differences), then the number of chimpanzees would need to be higher. Perhaps this should be explicitly discussed and highlighted when conclusions are drawn.

More specific comments regarding the validity of the findings / conclusions:

- Food preferences: perhaps the chimps' individual carrot/grape ratio preference could be included in the analysis. Overall, the chimps prefer grape, but the difference between 80/20 and 100/0 could weigh in a bit when putting the findings into perspective

- Line 380: Here, the question arose whether all chimps had experienced both locations throughout the 90 sessions. This is answered later in line 404, but could be addressed earlier because it can help put the concerning paragraph into perspective (the thought that crossed my mind was that if some or just one chimp(s) have only experienced the far location, the grapes, then maybe this explains their collective preference for going the distance (result in line 389)).

- Line 408-411: A little bit addressed later, but this could be taken as an indication against social learning, correct? I mean, perhaps this could be taken to balance the message / conclusion on yes/no social learning (also, more generally, in my view, the aspect of the necessity for chimpanzees to learn the new location socially rather than individually could be addressed a bit more, in light of their intelligence, explorative behavior and the fact that spatial differences do not comprise complex features to be learned. Agreed that local enhancement might be at play, and I think that the authors wield a perfectly adequate level of interpretation throughout their manuscript, but since chimps use many different ways of learning, and these chimps of course know their environment very well, and might explore it regularly, it could be interesting to discuss these aspects in relation to each other a bit more).

- Line 432-437: I assume that the authors have good reasons to propose this competition hypothesis, also because they might have observed these competition dynamics throughout the experiment, but apart from the result expressed in this sentence, CH has also more close than far exchanges in phase 1, while at least in token exchange numbers, there seems to be more competition at the close location. Also, in phase 3, CH has the highest number of far exchanges, while there is the most competition. Can the authors rhyme these results with their competition suggestion? Might there be more behavioral indications of competition?

- Line 470-472: The correlation does not necessarily imply this conclusion, are there more numbers to present? Or it could be phrased differently, if it's true that is of course: "HR more likely than LR individuals to scrounge food rewards rather than steal tokens".

- Line 532-534: Maybe not personality, but his role as alpha? Both tests were in the social group. Don't alpha males tend to participate less or less eagerly in general? But hmm, not anything substantial that can shed light on either of these hypotheses.

- Line 556: In light of CH's exchanges, I wouldn't use "necessity" here. And in the tables she's aged 13.

- Line 557-560: In light of her exchanges in all phases, I wouldn't necessarily conclude this, not without knowing more about her behavior for example.

- Line 582: More than the design (which also contributes I'm sure), in my view, the low number of individuals in the given space alleviates competition.

Additional comments

The study is refreshing and good because of its design simplicity and seemingly great execution. Very nice. I hope my comments can be helpful, especially in light of the sample size and the competition interpretation.

---

## Round 0.2 · Minor Revisions

Thank you for being so responsive to the comments from the reviewers. I have a few remaining points of clarification I would like you to consider before I can formally accept the manuscript, largely to improve the focus of the manuscript. I think the contributions can still be made more explicit in the abstract and introduction.

Delete the following sentence from the abstract, “The chimpanzees learned
how to exchange tokens for food rewards, but there was individual variation in the time it
took for them to do so”. It does not add much and the abstract is still quite lengthy. I also think the following sentence is unnecessary, “Furthermore, previous research has shown that dominants often scrounge food and tokens from their (less dominant) group mates. Although dominants did scrounge, and were more likely to scrounge for food rewards than tokens, levels of scrounging were low”. There is no real concluding statement, however. What is the take home message from your findings with regard to the theoretical impetus for the study?

Competition is introduced in the opening paragraph, but how is it tested here? On lines 279-282, wouldn’t the number of other chimpanzees present be at least as important as the number of visitors presented? This factor should be included in the analysis to consider the impact of competition, given your focus on competition in the introduction. If it is of interest that the chimpanzees exchanged at different locations within or across trials, what would the interesting factors be supporting this change in strategy? Here again, can you analyze as a function of number of competitors or types of competitors (dominant, subordinate) at different sites?

The introduction is improved but still reads as if the hypotheses are largely exploratory, e.g., “…we were interested in whether the chimpanzees would attempt to obtain tokens or rewards opportunistically, by scrounging them from their group mates…”. Can you make specific predictions for the chimpanzees’ behavior making reference to the variables of interest rather than simply stating what behaviors might be interesting to observe? For instance, if you assume that dominant animals will learn more quickly to transfer tokens, state this as an explicit hypothesis to be tested. I think the discussion does a better job of highlighting the interesting findings in light of prior research. Several intriguing findings, however, are left somewhat unexplained. Why is the dominant male so reluctant to exchange? Why did one male exchange only at the far location? I realize your explanations may be purely speculative but perhaps such speculation could point the way to further research in a future directions section?

You need to indicate which statistical tests were conducted, t’s should not be capitalized and should include degrees of freedom. Specify the type of t-test you used. Exactly what variables are being correlated on lines 302-303 – sessions/days? Why do you not report data from Phase 2 on lines 284-293?

And a few very specific edits:

On line 30, rewrite as “…they will detour from a direct path to food only when it enables…”
Likewise, on line 343, change “only represented” to “represented only”
On line 48 change “meant” to “means” so tense agrees.
Remove the extra “to” on line 105.

The addition of the supplemental video is very nice.
Thank you again for your work to improve the manuscript in light of the reviewers’ thoughtful commentaries,

---

## Round 0.3 · accepted · Accept

Thank you for being responsive to this second round of minor revisions. I am happy to accept the revised manuscript with the following corrections:

On line 50 change "offers" to "offer"
Should line 55 end with "foods"?
Edit the parentheses on lines 73-75.